# Fabrication of Polymer Micelles with Zwitterionic Shell and Biodegradable Core for Reductively Responsive Release of Doxorubicin

**DOI:** 10.3390/polym11061019

**Published:** 2019-06-09

**Authors:** Junting Jiang, Junbo Li, Biyu Zhou, Chaohuang Niu, Wendi Wang, Wenlan Wu, Ju Liang

**Affiliations:** 1School of Chemical Engineering & Pharmaceutics, Henan University of Science & Technology, 263# Kaiyuan Road, Luoyang 471023, China; kria1993129@163.com (J.J.); zhoubiyu1@163.com (B.Z.); nch19960430@163.com (C.N.); wang973180093@163.com (W.W.); liangju@haust.edu.cn (J.L.); 2School of Medicine, Henan University of Science & Technology, 263# Kaiyuan Road, Luoyang 471023, China; Whensa@sina.com

**Keywords:** poly-ε-caprolactone, polycarboxybetaine, triblock copolymer, disulfide-linker, anticancer drugs, release

## Abstract

To achieve a high stability in physiological environment and rapid intracellular drug release, a biodegradable zwitterionic triblock copolymer with a disulfide-linked poly-ε-caprolactone and polycarboxybetaine methacrylate (PCBMA-SS-PCL-SS-PCBMA) was prepared for micellar carrier to delivery doxorubicin (DOX) into tumor cells. PCBMA-SS-PCL-SS-PCBMA was obtained by following steps: i) introducing disulfide bonds through end-group modification of PCL diol with cystamine dihydrochloride; ii) preparing PCL-RAFT macromolecular chain transfer agent by EDC/NHS chemistry; iii) RAFT polymerization of zwitterionic monomer. Self-assembling from PCBMA-SS-PCL-SS-PCBMA, polymeric micelles had many advantages, such as ultra-low protein absorption in serum and obvious reduction-responsiveness in the presence of DTT. Furthermore, DOX-loaded micelles exhibited high stability upon centrifugation and lyophilization, a fast intracellular drug release and enhanced drug efficacy due to GSH-triggered PCBMA shell shedding and micellar reassembling. Thus, the polymeric micelles integrated several functions and properties could be prospectively utilized as valuable nanocarriers in cancer chemotherapeutics.

## 1. Introduction

During the past decades, development of nano drug delivery systems (NDDS) has received tremendous attention due to their altering pharmacokinetic profile and reducing side effects of hydrophobic antitumor drugs in cancer chemotherapeutics [1,2,3]. Typically, the ideal NDDS ought to possess the following characteristics: Efficient drug encapsulation, high stability in physiological environment, and rapid intracellular cargo release. Polymeric micelles self-assembled from amphiphilic copolymer, have emerged as a valid NDDS due to their flexible structure and tailored properties [4,5]. For instance, their core–shell architecture is particularly suitable for efficiently hydrophobic drug encapsulation and to form homogeneous suspension in vivo, resulting from their hydrophobic inner core and hydrophilic outer shells [6,7]. Moreover, their neutral hydrophilic shells effectively reduce interactions with blood components and avoid clearance by the reticuloendothelial system (RES), thereby prolonging blood circulation times [8,9]. Finally, loaded drugs accumulate at tumor site through the so-called enhanced permeation and retention (EPR) effect [10].

To date, Polyethylene glycol (PEG) is neutral hydrophilic polymer most used to prevent nonspecific protein adhesion on account of the formation of a hydration layer via hydrogen bond interaction [11,12]. However, PEG presents several limitations, such as easily suffering from oxidization and accelerated blood clearance (ABC) phenomenon caused by the generation of anti-PEG immunoglobulin antibodies [13]. Recently, zwitterionic polymers have emerged as a neutral ultra-hydrophilic hydrophilic polymer alternative to PEG [14,15]. Zwitterionic polymers bear equimolar number opposite ionizable groups. Poly(sulfobetaine methacrylate) (PSBMA) and poly(carboxybetaine methacrylate) (PCBMA) nanoparticles have been shown to have better blood compatibility and higher resistance to protein adsorption than PEGlated nanoparticles due to more tightly bound water molecules via ionic interactions [16,17]. For instance, Thus, zwitterionic nanoparticles can undergo repeated blood circulation without detectable immune response, leading to high accumulation at tumor sites.

In addition to high stability in physiological environment, an optimal micellar carrier should precisely control drug release inside the cancer cell. However, therapeutic efficacy is always limited by the gradual permeation of hydrophobic drugs from the hydrophobic core [18]. For instance, polymeric micelles with a biodegradable polyester core, such as poly(Ɛ-caprolactone) (PCL) have received much attention in past decades owing to safer application among synthetic polymers [10,19,20]. The biodegradation of PCL may accelerate drug exudation from hydrophobic core, mostly relying on enzymatic interfacial activation [21]. However, peripheral dense shells hinder the access of enzyme to PCL core, resulting in a slow degradation rate [21,22]. The development of stimuli-responsive materials and carriers [23,24,25,26,27] that quickly release encapsulated drugs triggered by intracellular environmental changes is an efficient strategy to improve drug bioavailability [28,29,30,31]. For instance, shell-sheddable biodegradable micelles responsive to glutathione (GSH) have been developed to take advantage of the large GSH concentration difference between the extracellular and intracellular environments [32,33]. These carriers showed faster intracellular release of doxorubicin (DOX) due to GSH-triggered clearance of intermediate disulfide bonds between shell and core, thereby allowing drug extrusion during micellar restruction.

Herein, an amphiphilic triblock copolymer with disulfide-linked poly-ε-caprolactone and polycarboxybetaine methacrylate (PCBMA-SS-PCL-SS-PCBMA) was used to fabricate polymeric micellar carriers with high resistance to protein adsorption and reductive responsive intracellular release. After encapsulating DOX, the anticancer effect of the carrier was investigated with hepatoblastoma cells line (HepG2). The process of forming DOX loaded PCBMA-SS-PCL-SS-PCBMA micelles and redox responsive intracellular release is illustrated in Scheme 1.

## 2. Experimental Section

### 2.1. Materials

PCL-diol (Mn = 2000, PDI = 1.2) was purchased from Sigma-Aldrich (Shanghai, China). Cystamine dihydrochloride (Cys.2HCl) (>96%), 1, 10-carbonyldiimidazole (CDI) (>97.0%) were purchased from National Pharmaceutical Group Chemical Reagent (Shanghai, China) and used as received. 2,2-Azobis(isobutyronitrile) (AIBN) (97%) was purchased from J&K Chemical (Beijing, China) and recrystallized before being used as an initiator. The monomers, Carboxybetaine methacrylate (CBMA, 99%), RAFT reagent (4-cyanopentanoic acid)-4-dithiobenzoate (CPAD, 98%), Doxorubicin hydrochloride (DOX·HCl, 99%), *N*-(3-Dimethylaminopropyl)-*N’*-ethyl carbodiimide hydrochloride (EDC·HCl, 98%) and *N*-Hydroxysuccinimide (NHS, 98%) were purchased from J&K Chemical, (Beijing, China) and used as received. All other reagents and solvents were of analytical grade and used as received. Dulbecco’s modified Eagle’s medium (DMEM), penicillin-streptomycin, fetal bovine serum (FBS) and 3-[4,5-dimethylthiazol-2-yl]-2,5-diphenyltetrazolium bromide (MTT), were purchased from Thermo Fisher Scientific (Shanghai, China).

### 2.2. Midifaction of PCL-Diol with Cystamine (Cys-PCL-Cys)

A solid mixture of PCL-diol-2000 (2.0 g, 1 mmol) and CDI (1.62 mg, 10 mmol) was added to a three-neck round bottom flask and dried under vacuum at 60 °C overnight. Next, 30 mL of dry dichloromethane and pyridine was quickly injected into the flask under nitrogen protection. The reaction was carried out at room temperature (RT) for 24 h under stirring. The product was concentrated and precipitated in excess cold diethyl ether. The solution–precipitation process was repeated three times to ensure removal of excess CDI. After vacuum drying, the structure of CDI-PCL-CDI (yield: 85%) was verified by ^1^HNMR spectroscopy for the following step.

To obtain PCL-cystamine conjugates, Cys.2HCl (1.36 g, 6 mmol) was first dissolved in 40 mL mixed solvent of dimethyl sulfoxide (DMSO) and pyridine (1:1 *v*/*v*) containing 2 mL trimethylamine (TEA). Next, this solution was added dropwise into a solution prepared by dissolving 1 g CDI-PCL-CDI into 25 mL DMSO and pyridine with a volume ratio of 1.5:1. The reaction was carried out at RT for 48 h with stirring under nitrogen protection. The product was purified with dialysis against deionized water to remove excess Cys.2HCl and solvents. Finally, the aqeous phase was lyophilized to collect PCL-cystamine conjugate (Cys-PCL-Cys, yield: 79%). The chemical structure was verified using ^1^HNMR spectroscopy (400 MHz, DMSO-d6): d (ppm) = 3.96–3.99 (t, –CH_2_–O–), 2.71–2.77 (m, –CH_2_S–), 2.25–2.31 (t, –CO–CH_2_–CH_2_), 1.51–1.56 (m, –CO–CH_2_–CH_2_–CH_2_ and CH_2_–CH_2_–CH_2_–O–), 1.29–1.36 (m, –CH_2_–CH_2_–CH_2_–).

### 2.3. Synthesis of CPADB-SS-PCL-SS-CPADB Macro-RAFT Agent.

CPADB (1.5 g, 4.6 mmol) and NHS (1.1 g, 9.4 mmol) were first mixed in 25 ml dry CH_2_Cl_2_ in a three neck round flask. Next, DCC (2.9 g, 14.1 mmol) was added into the flask with stirring under nitrogen protection. The reaction was allowed to proceed at RT for 48 h in the dark. After removing the insoluble salts by suction filtration, the filtrate was concentrated and further purified by chromatography (silicagel, hexane/ethyl acetate = 6/4, *v*/*v*). After removing all the solvents, CPADB-NHS was obtained as a dark-red solid (yield: 72.3%).

To synthesize PCL macromolecular RAFT reagent, Cys-PCL-Cys (0.6 g, 0.165 mmol) and CPADN-NHS (0.3 g, 0.67 mmo1) were first mixed in 15 mL dry CH_2_Cl_2_ and left at RT for 48 h in the dark. The solvent was then concentrated and precipitated in excess cold diethyl ether. After filtration and dry in vacuum, CPADN-SS-PCL-SS-CPADN was obtained as a light-red solid (yield: 62.9%).

### 2.4. Synthesis of PCBMA-SS-PCL-SS-PCBMA

The CPADB-SS-PCL-SS-CPADB macro-RAFT agent (0.2 g, 0.070 mmol) and AIBN (2.3 mg, 0.014 mmol) were dissolved in 4 mL dry THF in a sealed ampoule. Then zwitterionic monomer CBMA (0.3 g, 1.308 mmol) was first dissolved in 4 mL saturated saltwater and added to the ampoule above. After the solution was degassed by three freeze-evacuate-thaw cycles, the polymerization was conducted at 60 °C for 48 h. The product was purified by dialysis (bag MWCO = 3000) against deionized water to remove salt and unreacted CBMA. Finally, PCBMA-SS-PCL-SS-PCBMA was obtained by lyophilization as a light pink powder (0.23 g, 46.4%).

### 2.5. Preparation of PCBMA-SS-PCL-SS-PCBMA Micelles

PCBMA-SS-PCL-SS-PCBMA (10 mg) was first dissolved in DMSO (2 mL) to prepare polymer solution with an initial concentration of 5.0 mg/mL. Subsequently, PB (50 mM, pH 7.4) was added to the polymer solution at a rate of one drop every 6–7 s with vigorous stirring until the solution turned turbid, indicating the occurrence of micellization of triblock copolymers. The micelle solution was left overnight and then transferred into a dialysis bag (MWCO = 3500) against PB for 48 h to remove DMSO. Finally, the micelle solution was diluted to 0.5 mg/mL of polymer concentration.

The critical micelle concentration (CMC) was determined by using pyrene as a fluorescence probe on a Varian fluorescence spectrophotometer at RT. The concentration of PCBMA-SS-PCL-SS-PCBMA varied from 2 × 10^−4^ to 0.2 mg/mL in water and the concentration of pyrene was fixed at 0.6 μM. The fluorescence spectra were recorded using a FLS920 fluorescence spectrometer with the excitation wavelength of 330 nm. The emission fluorescence was monitored at 368 and 375 nm. The CMC was estimated as the cross-point when extrapolating the intensity ratio *I_3_*_75_/*I*_368_ at low and high concentration regions.

The size and size distribution (PDI) of micelles were estimated with a laser particle size analyzer (Zetasizer Nano, Malvern, UK). Micelle morphology was observed under a Hitachi H600 transmission electron microscopy (TEM) system at an operated voltage of 75 kV. For TEM measurement, the sample was prepared by adding a drop of micelle solution onto the copper grid, and then the sample was air-dried and measured at RT.

### 2.6. DOX Loading and Release

The stock solution of the hydrophobic drug DOX was firstly prepared by mixing DOX·HCl with TEA in dimethyl sulfoxide (DMSO) for 2 h. Subsequently, 2 mL DOX solution (1 mg/mL) was added to 10 mg PCBMA-SS-PCL-SS-PCBMA. After stirring overnight, PB (50 mM, pH 7.4) was added to the above solution at a rate of one drop every 6–7 s until the formation of PCBMA-SS-PCL-SS-PCBMA micelles was achieved. The solution was transferred into a dialysis bag (MWCO 3500) against PB for 48 h and then filtered through a 0.45 μm filter to remove the unloaded DOX. DOX-loaded micelles were lyophilized for the next experiment. For determination of drug loading content, lyophilized DOX-loaded micelles were dissolved in DMSO and analyzed with a UV-Vis spectrophotometer (absorbance intensity at 485 nm). Drug loading content (DLC) and drug loading efficiency (DLE) were calculated from the following equations:(1)$$\mathrm{DLC}\left(\%\right)=\frac{\mathrm{weight}\;\mathrm{of}\;\mathrm{loaded}\;\mathrm{drug}}{\mathrm{weight}\;\mathrm{of}\;\mathrm{drug}-\mathrm{loaded}\;\mathrm{micelles}\;}100\%$$
(2)$$\mathrm{DLE}\left(\%\right)=\frac{\mathrm{weight}\;\mathrm{of}\;\mathrm{loaded}\;\mathrm{drug}}{\mathrm{weight}\;\mathrm{of}\;\mathrm{drug}\;\mathrm{in}\;\mathrm{feed}\;}100\%$$

In vitro release profiles DOX from polymer micelles were investigated using a dialysis tube (MWCO = 14,000) at 37 °C against PB (50 mM, pH 7.4) and PB (50 mM, pH 5.0) with or without 10 mM DTT. At regular time intervals, 3 mL of release media was taken out and replaced with an equal volume of fresh media. The amount of DOX released was determined by UV-vis spectroscopy. The release experiments were conducted in triplicate. The results presented are the average data with standard deviations.

### 2.7. Cell Uptake Studies

The cellular uptake of DOX and DOX-loaded micelles was observed by fluorescence microscopy. HepG2 were seeded into 24-well plates at an initial density of 5∙10^4^ cells/well with 1 mL DMEM containing 10% FBS and incubated at 37 °C for 24 h in 5% CO_2_. After reaching about 80% confluence, the cells were incubated with free DOX, or DOX-loaded micelles. After incubation, the cells were washed three times with PBS and then observed by fluorescence microscopy. Next, the cells were detached with 0.25% trypsin and resuspended in 500 μl PBS (pH 7.4) for measurement with a flow cytometer (FC500, Beckman Coulter, Midland, ON, Canada).

### 2.8. Cell Viability Assays

The cytotoxic effects of empty micelles or DOX-loaded micelles were determined by using MTT assays. HepG2 was seeded into 96-well plates at 5000 cells/well and cultured 24 h in 200 μL DMEM containing 10% FBS. The culture medium was replaced with PBS (pH 7.4) containing free DOX, DOX-loaded micelles, and then the cells were incubated for 24 h. The medium was replaced with 200 μL of fresh medium and MTT (20 μL, 5 mg/mL in PBS) stock solution was then added to each well. After 4 h, unreacted dye was carefully removed and formazan crystals were dissolved in 200 μL/well DMSO. The plate was incubated for another 10 min before measuring absorbance at 570 nm with an ELISA microplate reader (Bio-Rad). Cell viability (%) was calculated as follows: Cell viability (%) = (OD_sample_/OD_control_)∙100, where OD_sample_ is the absorbance of the cells treated by polymers, and OD_control_ is the absorbance of the untreated cells. Each experiment was done in triplicate.

## 3. Results and Discussion

### 3.1. Polymer Synthesis and Characterization

Scheme 2 represents the synthesis route of PCBMA-SS-PCL-SS-PCBMA. Two PCL-diol hydroxyl groups activated by CDI were used to introduce a cystamine monomer, which provided reductive disulfide bonds and reactive amino-end groups [34]. Next, a PCL macro-RAFT agent was readily achieved through a conjugating reaction between amino-end groups of Cys-PCL-Cys and the carboxy group of CPADB through EDC/NHS chemistry. The zwitterionic monomer CBMA can typically polymerize in water or methanol, whereas the PCL macro-RAFT agent is not soluble in either of these solvents. Thus, we selected as solvent a mix of THF and saturated saltwater in equal volumes to ensure homogeneous RAFT polymerization. The PCL macro-RAFT agent and PCBMA-SS-PCL-SS-PCBMA were characterized by ^1^HNMR and FTIR spectroscopy.

As shown in Figure 1A, the ^1^HNMR spectrum of PCL macro-RAFT have new characteristic peaks at i, l, j, and k, which are attributed to the protons of CPADB, in addition to the characteristic protons of PCL at peaks c, d, e, and f (δ4.0, 2.3, 1.5, and 1.3 ppm). Subsequently, PCBMA-SS-PCL-SS-PCBMA was obtained by direct RAFT polymerization with PCL macro-RAFT and CBMA monomer. Compared with PCL macro-RAFT, the ^1^HNMR spectrum of PCBMA-SS-PCL-SS-PCBMA shows new peaks at t, u, p and q (δ4.6, 3.8, 3.6, 3.3 and 2.5 ppm), which belongs to the characteristic protons of betaine carboxylate. Here, the molecular weight of PCL was 2000. The unit number of CBMA is estimated by the integral area ratio of t and c, which was calculated as 11. Thus, the molecular weight of PCBMA-SS-PCL-SS-PCBMA is 6400 g mol^−1^. FT-IR spectrum was also used to determine the successful synthesis of PCL macro-RAFT and PCBMA-SS-PCL-SS-PCBMA. As shown in Figure 1B, the peak at 2958 cm^−1^ belongs to the methylene in repeated unit of PCL. The introduction of cystamine and CPADB at end groups of PCL is verified by peaks at around 1625 cm^−1^ and 1575 cm^−1^ of amide [35]. Finally, the presence of absorbance peaks at 1144 cm^−1^ is the vibration of C–N in quaternary ammonium of betaine carboxylate [36].

### 3.2. Characterization of Polymeric Micelles

PCBMA-SS-PCL-SS-PCBMA was used to prepare polymeric micelles in aqueous solution using PCL as core and PCBMA as shell. The critical micelle concentration (CMC) of PCBMA-SS-PCL-SS-PCBMA was determined by fluorescence spectroscopy using pyrene as a probe. Figure 2A shows the intensity ratio of pyrene emission spectra at 375 nm (I_3_) to 368 nm (I_1_) (I_3_/I_1_) as a function of the polymer concentration. The I_3_/I_1_ ratio increases significantly at a specific polymer concentration, indicating an obvious polarity change of the pyrene environment. The CMC value of PCBMA-SS-PCL-SS-PCBMA was estimated to be approximately 9.8 mg/L. A relatively low CMC is helpful to avoid the disassociation of micelles, which may occur as delivery drug systems become diluted in body fluids [37]. The micellar hydrodynamic diameter distribution (*D*_h_) was measured by DLS (Figure 2B). The average diameter is 87 ± 2 nm with a narrow size distribution from 61 to 89 nm. The polydispersity index (PDI) is approximately 0.1. TEM imaging revealed that the micelles have a spherical morphology with a mean diameter of approximately 70 ± 5 nm (inserted in Figure 2B). The difference between DLS and TEM measurements of micellar size is likely due to the shape of PCBMA at different hydration states, from swelled in aqueous medium (DLS) to collapsed at dry condition (TEM).

As the PCBMA shell links with the PCL core by disulfide bonds, polymer micelles may possess high antifouling and redox responsive properties. Serum stability of polymer micelle was first investigated by DLS analysis after incubation with PB containing 10% BSA and 50% FBS solution. As shown in Figure 3A, no significant changes in size were observed even after incubation for 72 h, showing that polymer micelles have high higher resistance to protein adsorption in these media. Redox response to GSH was further investigated by analyzing size change with DLS. The micelles showed increased *D*_h_ and broadened PDI after addition of 10 mM GSH for 4 h and 12 h. As disulfides are cleavable in the presence of GSH, the PCBMA shells were detached from the micellar surface, resulting in micelle aggregation.

### 3.3. Characterization of DOX Loaded Micelles

DOX was loaded onto PCBMA-SS-PCL-SS-PCBMA micelles by coprecipitation of PCL and DOX in PB. The incorporation of DOX into micelles was examined by fluorescence analysis (Figure 4A). A small shift in the absorption peak indicated an environmental change in DOX upon encapsulation into the micellar core [38]. Drug loading content was determined by UV-Vis spectrophotometry in DMSO. DLC and DLE were approximately 15% and 41%, respectively. The *D*_h_ of DOX-loaded micelles increased slightly to 124 ± 3 nm and maintained a narrow size distribution (Figure 4B), when compared to DOX-free micelles (Figure 2B). TEM imaging revealed that DOX-loaded micelles exhibit a spherical morphology with a diameter of approximately 102 ± 2 nm (Figure 4C). The stable structure and uniform nanoscale size of these carriers are fundamental properties for efficient intracellular uptake and EPR effect in vivo [39]. High-speed centrifugation or freeze-drying is a necessary procedure for the purification or storage of nanocarriers. The nanocarriers are always expected to retain high stability and uncompromising application effect after post processing. Jiang’s group verified that PLGA-PCB nanoparticles with PCB shell show a slight size change after repeated processing steps [20]. Here, DOX-loaded micelles were first centrifuged and lyophilized, then measured the resuspended size with DLS analysis. As shown in Figure 4D, the size of DOX-loaded micelles increases approximately only 5 ± 0.8 nm and 18 ± 1.2 nm after treatment, respectively. Compared with aggregated PEGylated nanoparticles upon similar post-formulation [40], the stability of zwitterionic micelles is due to stronger hydration layer via ionic interactions provided by the PCBMA shells [20].

### 3.4. In-Vitro Drug Release

In vitro drug release was conducted at 37 °C in PBS at pH 7.4 in the presence or absence of 10 mM DTT to mimic the intracellular microenvironment [41] (Figure 5). The reduction insensitive PEG-*b*-PCL micelles, with similar hydrophilic and hydrophobic chain length, were used to contrast the release behavior. From Figure 5, more than 34% of cargo released from PCBMA-SS-PCL-SS-PCBMA micelles within 4 h and approximately 61% after 48 h in the presence of 10 mM DTT. However, only approximately 31% of DOX was released within 48 h at pH 7.4 without DTT. A reductive environment triggered an accelerated release of DOX. In contrast, reduction insensitive PEG-*b*-PCL micelles show a similar release behavior in presence or absence of DTT, where only approximately 30% DOX was observed in 48 h. Under a reductive environment, cleavage of disulfide bonds between the PCBMA shell and the PCL core results in shell shedding and micelle aggregation. The drug was rapidly extruded during reassembly of deprotected micelles [33].

### 3.5. In Vitro Cytotoxicity and Cell Uptake

In vitro cytotoxicity of empty micelles and drug-loaded micelles was evaluated with MTT assays in HepG2 cells. Cells without treatment were used as control and showed a viability of 100%. As shown in Figure 6A, PCBMA-SS-PCL-SS-PCBMA and PEG-*b*-PCL micelles exhibited high biocompatibility with HepG2 cells at every concentration up to 200 mg/L. Low cytotoxicity was conferred to a good biocompatibility of both micelles. For the viability assays, HepG2 cells were incubated with free DOX, DOX-loaded PCBMA-SS-PCL-SS-PCBMA micelles, and DOX-loaded PEG-*b*-PCL micelles (Figure 6B). The inhibitory concentrations to produce 50% of cell death (IC50) in HepG2 cells were 0.26 mg/L, 0.72 mg/L, and 1.44 mg/L for free DOX, DOX-loaded PCBMA-SS-PCL-SS-PCBMA micelles, and DOX-loaded PEG-*b*-PCL, respectively. The PCBMA-SS-PCL-SS-PCBMA micelles showed significantly enhanced therapeutic efficiency than PEG-*b*-PCL micelles, which may be due to efficient intracellular release of DOX caused by GSH-triggered clearance of intermediate disulfide bonds.

DOX is a popular anticancer drug widely used in the treatment of various tumors. Mechanistically, the insertion of DNA into tumor cells inhibits macromolecular biosynthesis, eventually leading to cell apoptosis [42,43]. Intracellular drug levels depend on the efficacy of drug release from micelles [44]. The increased therapeutic effect of DOX-loaded PCBMA-SS-PCL-SS-PCBMA micelles in HepG2 cells can be attributed to improved intracellular release of DOX, as demonstrated by fluorescence microscopy (Figure 7A). Indeed, empty micelles had no fluorescence signal (Figure 7Aa), whereas the red fluorescence of DOX was detectable (Figure 7Ab,Ac), showing that DOX is efficiently delivered into HepG2 cells by both micellar carriers. Compared with Figure 7b, the intensity of red fluorescence was remarkably enhanced in the cytoplasm and nucleus, indicating effective and fast release DOX from PCBMA-SS-PCL-SS-PCBMA micelles. The flow cytometry was also used to measure the intracellular fluorescent signals. As shown in Figure 7B, HepG2 cells treated with PCBMA-SS-PCL-SS-PCBMA micelles exhibited about 3 times higher mean fluorescence intensity than those treated by DOX-loaded PEG-*b*-PCL micelles. These results are consistent with the toxicity studies described above.

## 4. Conclusions

In conclusion, a triblock copolymer with disulfide-linked zwitterionic and biodegradable polymer (PCBMA-SS-PCL-SS-PCBMA) was successfully synthesized through RAFT polymerization. PCBMA-SS-PCL-SS-PCBMA self-assembles into stable nanoscale micelles in aqueous solution, showing high antifouling ability and redox responsive property. DOX-loaded PCBMA-SS-PCL-SS-PCBMA micelles also exhibited an excellent stability upon post processing, rapid drug release under cellular reducing environment, and high antitumor activity to HepG2 cells. Thus, biodegradable zwitterionic polymeric micelles could be potential hydrophobic antitumor drug carriers for enhanced therapeutic efficacy.

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
