# Peer review of "Fabrication of Polymer Micelles with Zwitterionic Shell and Biodegradable Core for Reductively Responsive Release of Doxorubicin"

_polymers, 2019, doi:10.3390/polym11061019_

Reviewer 1 Report

Manuscript number polymers-493514, entitled “Fabrication Polymer Micelles with Zwitterionic Shell and Biodegradable Core for Redox Responsive Release of Doxorubicin” deals with the effective control of drug delivery using a biodegradable zwitterionic triblock copolymer with disulfide-linker. Although at first glance the manuscript looks very promising and data looks nice, there are many similar papers and the originality of this manuscript is not yet clear to me. For example, the content of Ref[16] is almost same as this manuscript. Also, there are many other papers such as ACS Appl. Mater. Interfaces, 2018, 10 (28), pp 23509–23521. The authors should list up the relevant papers and compare with them.

Author Response

Dear Referees,

Thank you very much for your effort and time devoted to our manuscript entitled “ Fabrication of Polymer Micelles with Zwitterionic Shell and Biodegradable Core for Reductively ox Responsive Release of Doxorubicin”. Your insightful comments greatly help us improving our manuscript. We have carefully studied all the comments and questions, then revised the manuscript accordingly. The detail point-by-point responses to the comments are attached in another file named “Authors’ Response to Reviewers’ Comments”. We sincerely hope that our revised manuscript is now fully satisfactory to warrant its publication in Polymers.

Sincerely,

Dr. Junbo Li

Reviewer 2 Report

Li and co-workers report on the preparation of triblock terpolymers consisting of poly-ε-caprolactone as middle and poly(carboxybetaine methacrylate) obtained after a disulfide linking reaction followed by RAFT polymerization. The polymers were designed with regard to their capability of micelle formation and loading of the anti-tumor reagent doxorubicin. The homopolymer and final block copolymers were investigated by state of the art polymer analytical tools comprising 1H NMR spectroscopy and FT-IR spectroscopy. Critical micelle concentration was determined via fluorescence spectroscopy intensities using pyrene as dye, while the presence of micelles were evidenced by transmission electron microscopy and dynamic light scattering. In general the manuscript needs English polishing by a native speaker and there are many misleading sentences, making it difficult to follow some of the assumptions. This manuscript could be suitable for MDPI Polymer after taking the following major aspects and suggestions into account (page numbers belong to the full pdf file sent):

Title: The title sounds strange to me: shouldn’t it be “Fabrication OF Micelles…”. Why is there an emphasis on redox-responsive, while reduction plays a major role (fully neglecting the reversibility of the investigated mechanism)?

Abstract: from my point the term “conjugated” is wrong in this particular case as it suggest an electron conjugation.

Introduction: The sentences “…due to their altered the pharmacokinetics…” and “Self-assembled form amphiphilic…” are wrong, i.e. do not make sense in a chemical way.

Within the Introduction the authors could add a section on oxidation- and reduction-responsive polymers for release studies and reviews in this field; Macromol. Chem. Phys. 2013, 214, 143; Journal of the American Chemical Society 2013, 135, 14198; ACS Nano 2012, 6, 9042; Macromolecular rapid communications 2016, 37, 1573; Macromolecules 2014, 47, 4876.

Page 2, Introduction: Please correct the following sentence: „… PEG is neutral hydrophilic…“

Page 2: please rephrase „ultrahydrophilic hydrophilic“

Page 2: within the second last section the following review given by Lowe and co-workers might be useful: Drug. Discov. Today 2019, 24, 129.

Scheme 1: please avoid the greyish background.

Scheme 1: the PCBMA structure is wrong: an acrylate is shown instead of a methacrylate

 Experimental Section: there are many typos, wrong indices, missing space characters, N in nitrogene-containing compounds should be italic; there should be a spectroscopy after 1H NMR,

Page 6: please correct “… which assigns to characteristic…”

Figure 1: resolution of the NMR spectra is poor (Fig 1A), therefore signal assignment is not clear.

There should be molar masses given for all prepared polymers as well as a polydispersity index value. This property directly correlates to the micellar characteristics and,moreover , is important for reproducing the obtained results.

Thermal investigations of the block copolymers can also be used to proof the existing block-like structure, but SEC is more meaningful in this case.

Section 3.2: please give statistics for all micelle sizes determined by DLS and TEM.

Page 6, line 232: rephrase “… from swelled at hydrated…”.

Figure 2: Please give the selective solvent within the legend.

Figure 3: The results, as determined by DLS measurements (Fig. 3B) do not reflect the sizes in dependency to the time of treatment (Fig. 3A). The significant change of PDI and size cannot be reflected by the minor changes of the statistics given in Fig. 3A. is the intensity of Fig. 3B based on weight intensity or amount of detected scattering? This point seems unclear.

Page 8: please give statistics for the loaded micelles.

Figure 5: space character missing in legend

Figure 7: resolution of legend in Fig. 7 B should be improved.

Author Response

Dear Referees,

Thank you very much for your effort and time devoted to our manuscript entitled “ Fabrication of Polymer Micelles with Zwitterionic Shell and Biodegradable Core for Reductively ox Responsive Release of Doxorubicin”. Your insightful comments greatly help us improving our manuscript. We have carefully studied all the comments and questions, then revised the manuscript accordingly. The detail point-by-point responses to the comments are attached in another file named “Response to Reviewers’ Comments”. We sincerely hope that our revised manuscript is now fully satisfactory to warrant its publication in Polymers.

Sincerely,

Dr. Junbo Li

Round  2

Reviewer 1 Report

The authors have added some data according to my suggestions. So now the manuscript has been improved for publication.

Reviewer 2 Report

The authors carefully addressed all raised aspects made by the reviewers. From my point of view this work is acceptable for publication in MDPI Polymers.